# Joint Modeling of Genetics and Field Variation in Plant Breeding Trials Using Relationship and Different Spatial Methods: A Simulation Study of Accuracy and Bias

**Éder David Borges da Silva** [1,2,]*, **Alencar Xavier** [3,4] **and Marcos Ventura Faria** [2]

[1] Department of Plant Breeding, Corteva Agriscience, Rio Verde Research Station, Rio Verde 75909-796, GO, Brazil

[2] Departamento de Agronomia, Universidade Estadual do Centro Oeste, Guarapuava 85040-167, PR, Brazil; mfaria@unicentro.br

[3] Department of Biostatistics, Corteva Agriscience, Johnston, IA 50131, USA; alencar.xavier@corteva.com

[4] Department of Agronomy, Purdue University, West Lafayette, IN 47907, USA

* Correspondence: eder.silva@corteva.com

**Abstract:** Modelling field spatial patterns is standard practice for the analysis of plant breeding. Jointly fitting the genetic relationship among individuals and spatial information enables better separability between the variance due to genetics and field variation. This study aims to quantify the accuracy and bias of estimative parameters using different approaches. We contrasted three settings for the genetic term: no relationship (I), pedigree relationship (A), and genomic relationship (G); and a set of approaches for the spatial variation: no-spatial (NS), moving average covariate (MA), row-column adjustment (RC), autoregressive AR1 × AR1 (AR), spatial stochastic partial differential equations, or SPDE (SD), nearest neighbor graph (NG), and Gaussian kernel (GK). Simulations were set to represent soybean field trials at $F_{2:4}$ generation. Heritability was sampled from a uniform distribution U(0,1). The simulated residual-to-spatial ratio between residual variance and spatial variance (Ve:Vs) ranged from 9:1 to 1:9. Experimental settings were conducted under an augmented block design with the systematic distribution of checks accounting for 10% of the plots. Relationship information had a substantial impact on the accuracy of the genetic values (G > A > I) and contributed to the accuracy of spatial effects (30.63–42.27% improvement). Spatial models were ranked based on an improvement to the accuracy of estimative of genetic effects as SD ≥ GK ≥ AR ≥ NG ≥ MA > RC ≥ NS, and to the accuracy of estimative of spatial effects as GK ≥ SD ≥ NG > AR ≥ MA > RC. Estimates of genetic and spatial variance were generally biased downwards, whereas residual variances were biased upwards. The advent of relationship information reduced the bias of all variance components. Spatial methods SD, AR, and GK provided the least biased estimates of spatial and residual variance.

**Keywords:** field plot variation; spatial adjustment; genomic selections; soybean breeding; spatial modeling

## 1. Introduction

Commercial breeding programs rely extensively on data-driven tools to efficiently improve crop genetics. The data analysis is based on modeling incorporate spatial and genetic relationship information.

In terms of modeling spatial patterns, plant breeding trials have historically utilized moving averages based on neighbor plots [1,2] as a simple method to account for within-field variation at higher resolutions than what is captured by the experimental blocks [3,4]. Spatial covariates have also been used in combination with other methods to increase accuracy and decrease bias [5,6], and it has shown favorable results when paired with genomic analysis [7]. Nevertheless, the spatial approaches have shifted from covariates to covariances with the advent of computing power, in which case the within-field variation

is reparametrized as structured random effects [8]. Under this framework, kernel methods, such as kriging and splines [9], are incorporated seamlessly into mixed models for plant breeding [10].

The current gold standard for the parameterization of spatial patterns in breeding trials is the two-dimensional auto-regressive structure referred to as AR1 $\times$ AR1 [11], where the dimensional correlations $(\rho_x, \rho_y)$ are estimated from data via restricted maximum likelihood [11–17]. AR1 $\times$ AR1 has served as a complementary approach to common experimental design [18], however, convergence can be a major issue due to the multiplicity of parameters being estimated [17]. Two kriging-type Matérn processes [19,20] that can be utilized as stable alternatives to AR1 $\times$ AR1 are the stationary Gaussian kernels [21], and the stochastics partial differential equation (SPDE), which correspond to a spline-like generalization of the AR1 process [22,23]. Yet, literature is scarce on the direct comparison among the multiple geostatistical methods when it comes to the analysis of plant breeding trials.

In terms of genetic information, modeling relationships are known to increase the accuracy of estimative genetic effects [24], with more impact in case of no replication or a few numbers of replications [25–29]. In addition, the joint model of genetic relationship has been reported to improve the signal separability between genetic and spatial variation [7,30,31].

The benefits in terms of accuracy attributed to relationship, spatial adjustment, and the combination of both have not been benchmarked, and their benefits remain unknown for early breeding stages. The goal of this study is to assess the accuracy and bias gains while comparing the combination of different parameterization approaches for the genetics and spatial patterns using simulation. Accuracy was measured on the coefficients as the correlation between estimated and true simulated values, and bias was measured on the variance components as the difference between estimated and true simulated values. Our simulation settings aim to reproduce the non-replicated trials from the early stages of varietal breeding programs, where the control of field plot variation is critical.

## 2. Material and Methods

### 2.1. Simulated Data

This section describes the general simulation settings, including the simulated genome, creation of the founder population and breeding parents, the experimental design, and simulated variance components. An overall representation of the simulation settings is provided in Figure 1. The next sections are reserved for the description of statistical models and the implementation details.

#### 2.1.1. Breeding Scenario

Simulation settings recreated early generation trials of a varietal breeding program, using soybeans (*Glycine max* L.) as template species to recreate the genomic settings. Soybean was chosen for its commercial importance [32]. Field settings were based on a single location containing balanced families, non-replicated entries, and performance checks. Entries were simulated to be at $F_{2:4}$ generation, where selection would be performed at a single-observation level.

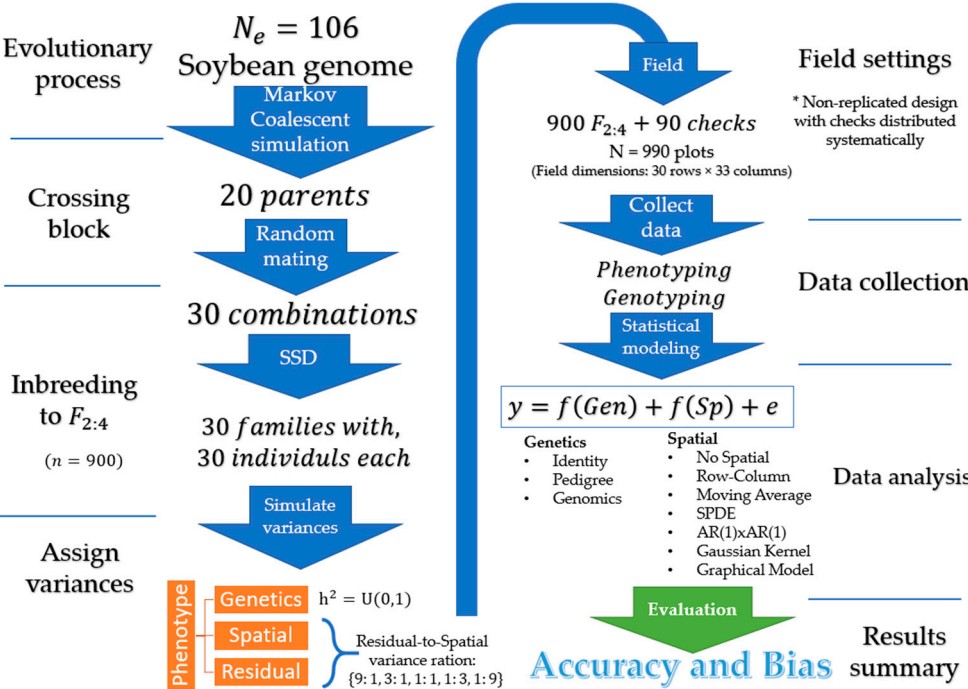

**Figure 1.** Illustration of the simulation steps: Create founder germplasm, form breeding population, assign variances, allocate plots into the field, data observation, statistical analysis, a summary of accuracy and bias.

### 2.1.2. Genome

Simulated genomic parameters were based on a soybean genome [33]. The genome contained 20 chromosomes, with an average length of 128 cM, covering and 950 Mb in genome size. The simulated genome contained 46,000 segregating polymorphic sites.

### 2.1.3. Breeding Lines

A set of founder lines was set with an effective population size ($N_e$) of 106 [34,35]. The founder lines would represent the entire diversity of the species, not the breeding program. From the 106 founders, we recreated an evolutionary process from which 20 breeding lines were produced. The evolutionary process was based on the Markovian Coalescent Simulator [36] that recreates multiple cycles of drift, mutation, and selection. The 20 breeding lines were utilized as parents for the populations under evaluation.

### 2.1.4. Breeding Trial

From the 20 breeding parents, 30 cross-combinations were performed at random. The 30 biparental families had 30 individuals each, totaling 900 breeding lines. The breeding trial included the 900 breeding lines in addition to 90 check plots, composed of 3 genotypes with 30 replications each. These were allocated into a rectangular field (30 rows × 33 columns) under a non-replicated augmented design, with checks systematically placed in the field [37]. Breeding lines, checks, and field layout were simulated separately in each iteration.

### 2.1.5. Genotyping and Breeding Values

Breeding lines were simulated and genotyped with 20,000 markers to mimic commercial SNP arrays available for soybeans, hence capturing a subset of all simulated polymorphic sites (46,000). True breeding values were obtained from the true genomic information under an infinitesimal model, hence all polymorphic sites had a small value sampled from a normal distribution. The observed SNP data would not include all true segregating loci, reproducing previous methodological frameworks [38]. The true breeding values were subsequently rescaled to display the variance parameters described next.

### 2.1.6. Population Parameters

A different set of variances were sampled for each run. First, a heritability value was sampled from a uniform distribution as $h^2 \sim U(0,1)$, from which the ratio between genetic variance $(\sigma_g^2)$ and residual variance $(\sigma_e^2)$ was estimated as follows: $\sigma_e^2 = (1 - h^2) \div h^2$, assuming $\sigma_g^2 = 1$. Spatial variance $(\sigma_s^2)$ was computed as a proportion of the residual variance, following the ratios: $\delta(\sigma_s^2 : \sigma_e^2) = $ 1:9, 1:3, 1:1, 3:1 and 9 : 1, thus $\sigma_s^2 = \sigma_e^2 \delta$. In terms of intra-class correlation coefficients, ICC $= \sigma_s^2 \div (\sigma_s^2 + \sigma_e^2)$, the rates correspond to ICC = 0.10, 0.25, 0.50, 0.75, and 0.90. The fixed ratios $\delta$ enables contrasting scenarios where the pure error is prevalent over spatial signal, and vice-versa. Variance components were subsequently rescaled to add up to 1.

### 2.1.7. Spatial Variation

Field spatial patterns were simulated from a stationary Matérn covariance function [39]. This function is commonly utilized for the simulation of spatial gradients in two-dimensional projections [40–42]. The signal from Matérn processes can be adequately recovered from most spatial procedures such as Gaussian kernels, splines, and auto-regressive moving averages [43]. A simulated field is presented in Figure 2.

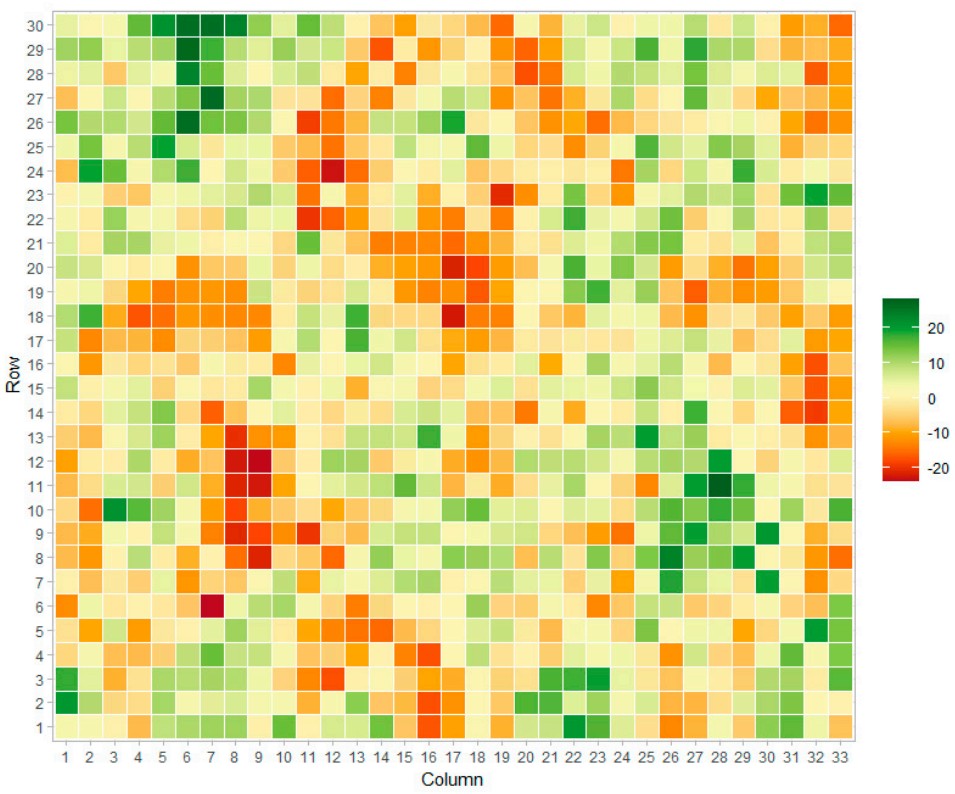

**Figure 2.** Example of simulated field signal using the Matérn function.

## 2.2. *True and Statistical Model*

### 2.2.1. True Model

The true model utilized to build the response (y) was based on a random effect model set as the linear combination of an overall mean (μ), genetics (g), spatial effect (s), and i.i.d. residuals (e). Thus,

$$y = \mu + g + s + e$$

Genetic, spatial, and residual terms were assumed to be normally distributed as $g \sim N(0, \mathbf{G}\sigma_a^2)$, $s \sim N(0, \mathbf{S}\sigma_a^2)$ and $e \sim N(0, \mathbf{I}\sigma_e^2)$. The covariance matrices correspond to the genomic relationship build from genome-wide polymorphism (**G**) and the Matérn covariance structure (**S**) described above. For simplicity, the true genetic term did not include dominance or epistasis effects.

### 2.2.2. Statistical Model

The statistical model consists of the parametrizations that attempt to recover the true simulated values. The technical details of the model parametrizations are described in the next two sections.

The notation "hat" represents the estimated coefficients and matrices, whereas the non-hat regards the simulated (or true) values. The genetic term was parameterized as unknown ($\mathbf{K} = \mathbf{I}$), based on single-generation pedigree information ($\mathbf{K} = \mathbf{A}$), and based on the estimated genomic information ($\mathbf{K} = \hat{\mathbf{G}}$) observed from the SNP array. The spatial term was set one of the following parametrizations: no spatial adjustment (NS), row-column adjustment (RC), moving average covariate (MA), AR1 $\times$ AR1 structure (AR), SPDE structure (SD), nearest neighbor graph (GM), or Gaussian kernel (GK).

### 2.2.3. Accuracy and Bias

The accuracy of coefficients is defined as the Pearson correlation between the estimate and true parameters. Hence, for the vector of genetic and spatial coeffects, the accuracy is defined as $\mathrm{cor}(g, \hat{g})$ and $\mathrm{cor}(s, \hat{s})$. The accuracy of selection for the top 10% estimated as the Jaccard coefficient of coincidence, measuring the percentage overall between the top-ranked individuals from the estimate and true breeding values. Bias was measured as the difference between the estimate and true parameters (i.e., $\hat{\theta} - \theta$). Measures of bias were computed for the variance components of the genetic, spatial, and residual terms.

### 2.3. Genetic Parametrizations

The genetic term of our statistical model was set as a random effect. Three parametrizations of the correlation matrix ($\mathbf{K}$) were:

### 2.3.1. No Relationship (K = I)

This parametrization is based on unstructured levels, which translated into the assumption that the relationship among genotypes is unknown. Under this setting, the genetic variance estimates largely rely on the replicated checks.

### 2.3.2. Pedigree Relationship (K = A)

In this study, the pedigree relationship matrix ($\mathbf{A}$) was based on a single generation, hence capturing relationship at the full- and half-sibling level. Such parametrization is well known in animal breeding and it is computationally efficient, since the sparse precision ($\mathbf{A}^{-1}$), the matrix can be directly computed [44].

### 2.3.3. Genomic Relationship (K = Ĝ)

The genomic relationship matrix [28] was computed as $\hat{\mathbf{G}} = \alpha \mathbf{M}\mathbf{M}'$, where $\mathbf{M}$ is the genotypic matrix containing individuals as rows and markers as column, coded as {0,1,2} then centered column-wise, and $\alpha$ is the normalizing factor computed as the sum of marker variance under Hardy–Weinberg equilibrium.

### 2.4. Spatial Parametrizations

The spatial term of our statistical model ($\hat{s}$) was parametrized in multiple ways with the purpose of benchmarking different strategies. These included:

### 2.4.1. No Spatial Term

This baseline parametrization sets $s_{NS} = 0$. Not adding a spatial term is herein informative to contrast the effect of adding spatial parametrization on the accuracy of coefficients and bias of variance components.

### 2.4.2. Row-Colum Effect

Two random effects, parametrizing each row and column as an independent level. This parametrization captures major linear patterns but no intra-field spatial gradients. The row-column term is described as $s_{RC} = s_{row} + s_{col}$, assuming normality as $s_{row} \sim N(0, I\sigma^2_{row})$ and $s_{col} \sim N(0, I\sigma^2_{col})$.

### 2.4.3. Moving Average Covariate

The moving average is a fixed effect covariate with one degree of freedom, $s_{MA} = x\beta$, also referred to as Papadakis covariate or Nearest Neighbors covariate [7,45]. The covariate is computed for each observation by average the observed phenotypic values of surrounding plots. We averaged the values of phenotypes within the distance of two rows and two columns.

### 2.4.4. AR1 × AR1

The two-dimensional autoregressive model AR1 × AR1 term is here described as $s_{AR} \sim N(0, \mathbf{R})$, where the spatial structure is dictated by $\mathbf{R} = (\Sigma_r \otimes \Sigma_c)\sigma^2_{AR}$, and $\Sigma$ represents the autoregressive matrices that specify the correlation ($\rho$) among levels. Two key properties of this method are: (1) involves estimating three parameters ($\sigma^2_{AR}$, $\rho_r$, $\rho_c$) via maximum likelihood and (2) the precision matrix ($\mathbf{R}^{-1}$) is sparse and can be computed directly as without building $\mathbf{R}$ [13,17,46].

### 2.4.5. Stochastic Partial Differential Equations (SPDE)

Under this framework, the spatial term is defined as $s_{SD} \sim N(0, \mathbf{B})$, where $\mathbf{B}$ is a spatial covariance structure build via SPDE [23,47], defined by the function $f(k) = (\kappa^2 - \Delta)^{0.5\,\alpha} x(s)$, where $\kappa$ is scale parameter, $\Delta$ is the Laplacian ($\sum_i \partial^2 / \partial k_i^2$), and $\alpha$ is a smother parameter ($\alpha = \nu + 0.5\gamma$) with dimension domain $\gamma$.

### 2.4.6. Nearest Neighbor Graph

This parametrization generalizes the moving average covariate into a dynamic nearest neighbor function [48,49] using the adjacent matrix (**H**) as design matrix of random effects, thus $s_{GM} = \mathbf{H}\gamma$ and $\gamma \sim N(0, I\sigma^2_\gamma)$. The matrix **H** was set to connect observations within two rows and two columns.

### 2.4.7. Gaussian Kernel

This uses a stationary Gaussian kernel to parametrize the covariance matrix of the spatial term [50]. In this method $s_{GK} \sim N(0, \mathbf{U}\sigma^2_{GK})$, where $\mathbf{U} = \exp(\alpha \mathbf{D}^2)$, as **D** is the Euclidean distance among plots based on row and columns, and $\alpha$ is a normalizing factor set ad hoc as 0.25.

### 2.5. Computation

The package AlphaSimR [38,51] was utilized to simulate the founder parents, to simulated the genome, to execute the Markovian coalescent simulator that generated the breeding parents, to simulate the crosses, to simulate the genotyping array, and to simulate the true breeding value.

Models were fit under the hierarchical Bayesian framework. Parameters were computed through Integrated Nested Laplace Approximation (INLA) to avoid computing Markov Chain Monte Carlo [22,52] with the R package INLA [47,52,53].

Covariance structures for SPDE and AR1 × AR1 are available in the INLA package [47]. The moving average covariate and the graphical model design matrix were based on the functions SPC and SPM, respectively, implemented in the bWGR package [54]. The genomic and pedigree relationship matrices, as well as the Gaussian kernel, were built using native R functions.

Subsequent statistical analyses of results were performed using the software R [55]. The code was run in parallel, distributed over 960 cores using the package doParallel [56].

Each scenario ($\delta$) was simulated 2000 times, computing all combinations of genetic and spatial parametrization.

## 3. Results

Accuracies and biases are summarized in Table 1 and Figures 3 and 4. Results indicate that adding relationship information has a major impact on the accuracy of genetic coefficients, whereas spatial adjustment was beneficial when the residual-to-spatial ratio is 1:1 or above. The least biased variance components were provided by SPDE, and methods that provided poor spatial adjustment led to upper biased residual variance.

**Table 1.** Marginal treatment averages and standard error across residual-to-spatial variance ratios.

| | | Accuracy | | | Bias | | |
|---|---|---|---|---|---|---|---|
| | | $cor(g,\hat{g})$ | $cor(s,\hat{s})$ | **Selection** | $\hat{\sigma}_g^2 - \sigma_g^2$ | $\hat{\sigma}_s^2 - \sigma_s^2$ | $\hat{\sigma}_e^2 - \sigma_e^2$ |
| Relationship | I | 0.569 (0.015) | 0.662 (0.039) | 0.395 (0.010) | −0.050 (0.004) | −0.146 (0.036) | 0.185 (0.035) |
| | A | 0.739 (0.009) | 0.683 (0.038) | 0.516 (0.009) | −0.005 (0.001) | −0.141 (0.036) | 0.142 (0.035) |
| | G | 0.782 (0.009) | 0.692 (0.037) | 0.562 (0.009) | −0.012 (0.001) | −0.134 (0.037) | 0.141 (0.035) |
| Spatial | NS | 0.661 (0.036) | - | 0.463 (0.027) | −0.030 (0.006) | - | 0.374 (0.069) |
| | RC | 0.661 (0.036) | 0.394 (0.025) | 0.463 (0.027) | −0.034 (0.007) | −0.362 (0.070) | 0.377 (0.069) |
| | MA | 0.705 (0.028) | 0.672 (0.034) | 0.497 (0.022) | −0.020 (0.005) | - | 0.155 (0.021) |
| | NG | 0.708 (0.027) | 0.765 (0.034) | 0.499 (0.022) | −0.024 (0.007) | −0.120 (0.025) | 0.076 (0.009) |
| | SD | 0.717 (0.025) | 0.778 (0.039) | 0.509 (0.020) | −0.007 (0.005) | −0.004 (0.008) | 0.007 (0.004) |
| | AR | 0.711 (0.026) | 0.678 (0.065) | 0.502 (0.021) | −0.018 (0.006) | −0.101 (0.015) | 0.046 (0.012) |
| | GK | 0.714 (0.027) | 0.786 (0.037) | 0.505 (0.022) | −0.023 (0.006) | −0.114 (0.021) | 0.057 (0.011) |

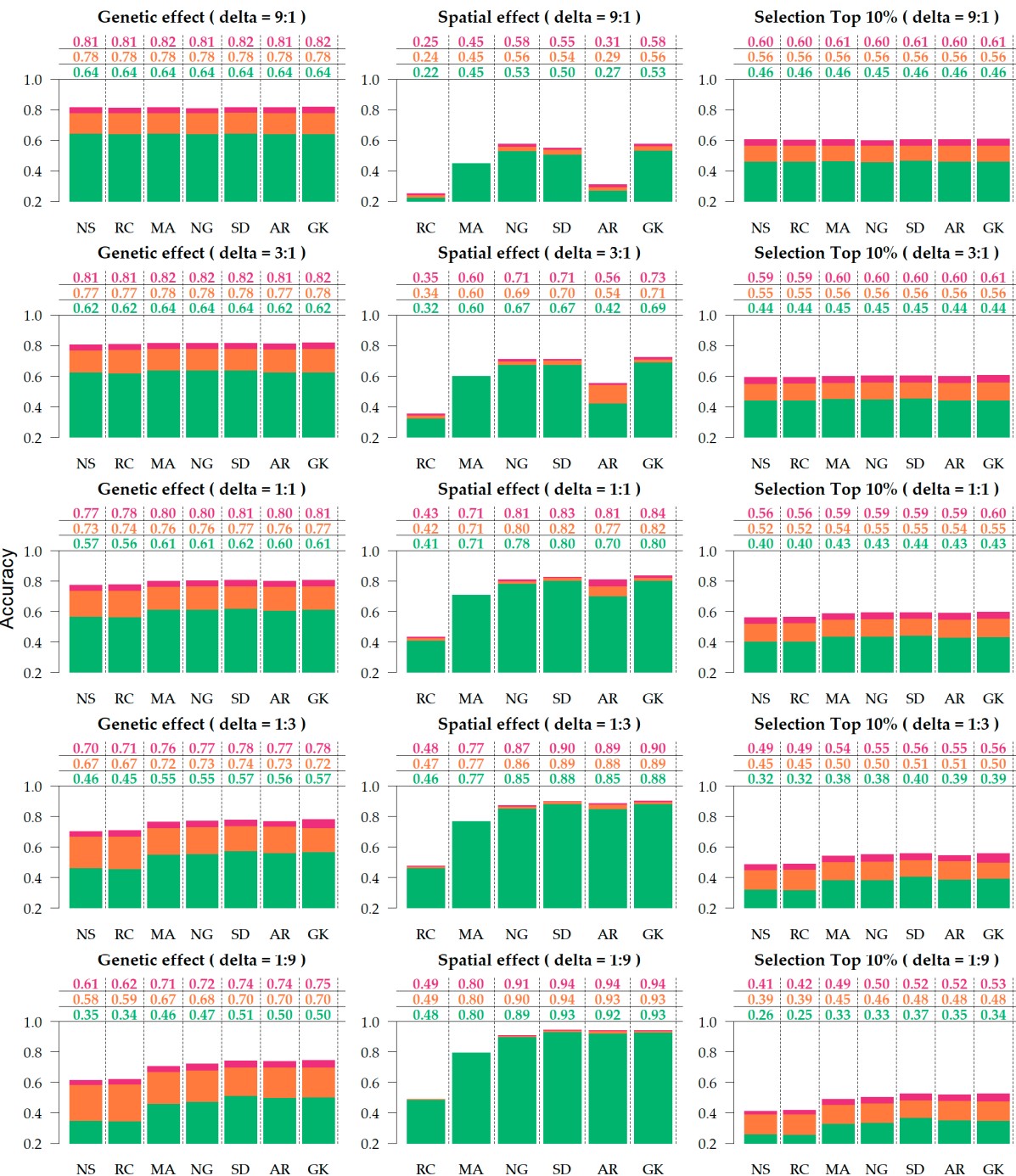

**Figure 3.** Accuracy of genetic effect, spatial effect, and selection, under varying residual-to-spatial variance ratios ($\delta = \sigma_e^2 : \sigma_s^2$). Colors correspond to the genetic relationship: Identity (green), pedigree (orange), genomics (red). Spatial adjustment methods on x-axis: no spatial adjustment (NS), row-column (RC), moving average covariate (MA), autoregressive AR1 $\times$ AR1 (AR), SPDE (SD), nearest neighbor graph (NG), and Gaussian kernel (GK).

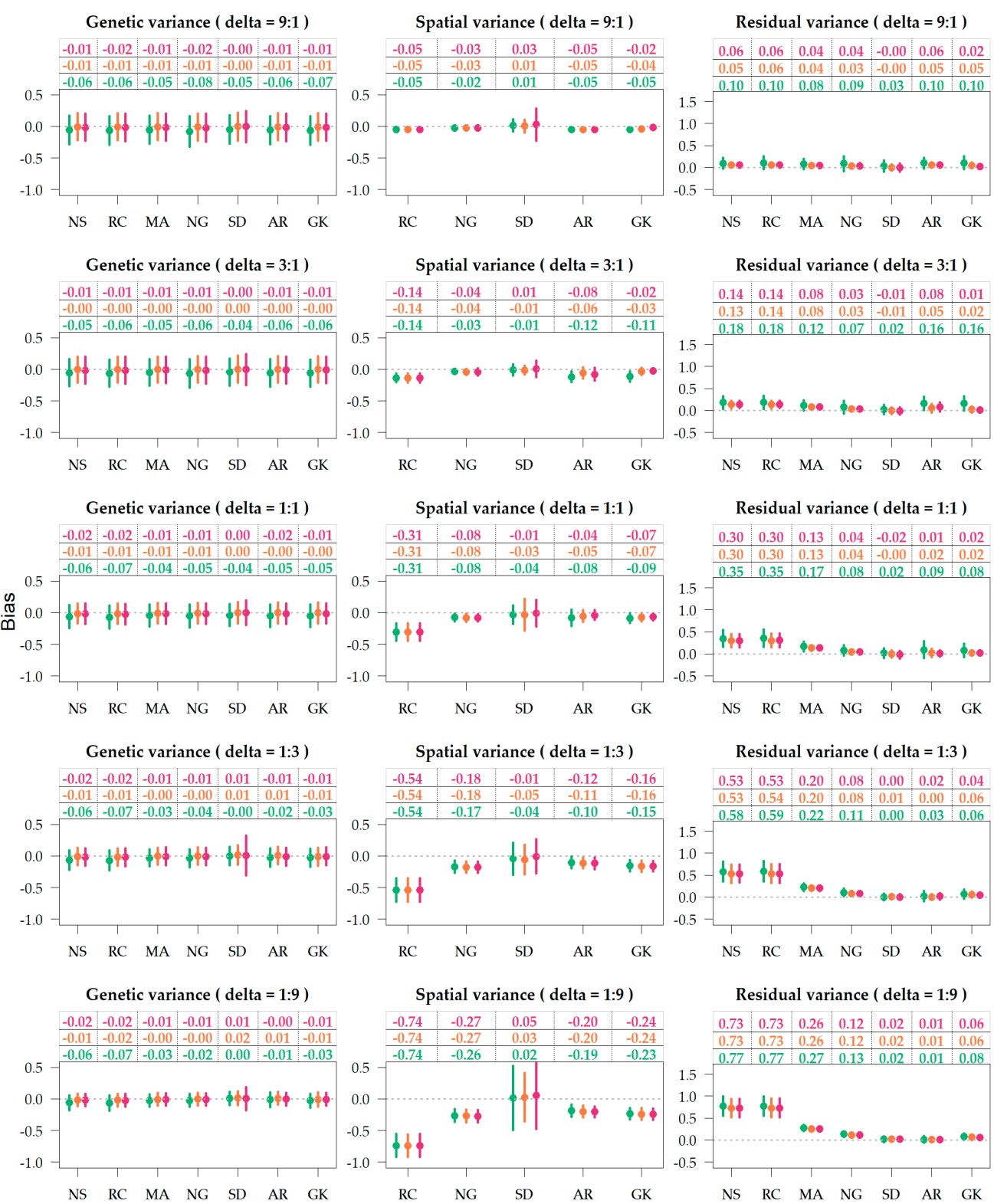

**Figure 4.** Variance bias of genetic, spatial and residual terms, under varying residual-to-spatial variance ratios ($\delta = \sigma_e^2 : \sigma_s^2$). Colors correspond to the genetic relationship: Identity (green), pedigree (orange), genomics (red). Spatial adjustment methods on *x*-axis: no spatial adjustment (NS), row-column (RC), moving average covariate (MA), autoregressive AR1 × AR1 (AR), SPDE (SD), nearest neighbor graph (NG), and Gaussian kernel (GK).

Across the various scenarios, the accuracy of genetic effects varied from 0.569 to 0.782. Based on the marginal averages (Table 1), the genetic accuracy increased from no relation-

ship (0.569) to a pedigree-based relationship (0.739) and increased further with genomic information (0.782). With regards to the contribution of the spatial term to genetic accuracy, the RC was equivalent to NS (0.661), whereas all other methods (0.705–0.717) provided an improvement on estimate genetic accuracy of approximately 0.05. The performance of spatial methods in the accuracy of genetic effects can be summarized as follows: SD $\geq$ GK $\geq$ AR $\geq$ NG $\geq$ MA > RC $\geq$ NS.

Higher gains in genetic accuracy were observed when adding relationships to NS, as it goes from no relationship (0.525) to pedigree (0.709) and genomics (0.749), increasing 0.184 and 0.224 in genetic accuracy, respectively, (Supplementary Tables S1 and S2) when compared to models with sophisticated spatial adjustment, such as SD, which goes from no relationship (0.598) to pedigree (0.756) and genomics (0.799) where the average gains were 0.158 and 0.201 for pedigree and genomic relationship, respectively (Supplementary Table S2). The benefits of this relationship also increase as the residual-to-spatial ratio decreased, as the average improvement of pedigree over no relationship is 0.136 at $\delta = 9 : 1$ and 0.170 when $\delta = 1 : 9$. Similarly, the improvements attributed to the genomic relationship were 0.176 at $\delta = 9 : 1$ and 0.213 at $\delta = 1 : 9$ (Supplementary Table S1).

Accuracies of spatial effects varied from 0.241 to 0.940 across simulated scenarios (Supplementary Table S3). The accuracy of spatial effects increased with spatial variance increase, such that little to no gain is observed when the residual-to-spatial variance ratio is 9:1 and 3:1, and negligible gains when the ratio is 1:1 (Figure 3). The contribution of adding a relationship to the spatial accuracy was small, (Supplementary Table S4), but beneficial. It provided a marginal increase from 0.021 and 0.030, going from no relationship (0.662) to pedigree (0.683) and genomic (0.692) relationship (Table 1).

Figure 3 shows that SD, AR, and GK provided the most accurate spatial predictions for scenarios where the spatial signal is greater than the random noise (1:3 and 1:9). In noisy scenarios (9:1 and 3:1), AR1 $\times$ AR1 did not perform as well, and the methods that provided the most accurate predictions were SD, GK, and the NG. Based on the marginal performance of accuracy in spatial effects (Table 1), the models can be ranked as follows: GK $\geq$ SD $\geq$ NG > AR $\geq$ MA > RC.

The accuracy of selection of the top 10% ranked from 0.306 to 0.607 (Supplementary Table S5). This metric provided similar results to those observed on genetic accuracy since the accuracy to selection is an alternative measure of the accuracy of genetic effects with specific application to breeding purposes. The advent of relationships provided a great impact on selection (Table 1 and Supplementary Table S6), with an improvement from 0.395 without a relationship, to 0.516 and 0.562 when using pedigree and genomic relationship information, respectively, with the corresponding enhancement of 30.63% and 42.27%. The different spatial parameterizations also differed with regards to their contribution to selection accuracy, particularly on scenarios where the spatial signal was strongest (1:9 and 1:3). Figure 3 shows that in the extreme case (1:9), the enhancement from NS to SD is 0.120, both with genomic relationship, from 0.410 to 0.530 and without relationship 0.080, from 0.260 to 0.340.

The biases of the variance components are illustrated in Figure 4, and the marginal averages across delta scenarios are provided in Table 1. Supplementary Tables S3 and S4 present the marginal averages for the different combinations of relationship, spatial method, and delta. For all purposes, bias values closer to zero are desirable and an estimator is considered unbiased if the bias is exactly zero. Herein, bias indicates if certain parametrizations are under-or over-estimating the variances attributed to the different model terms. For the methods NS and MA, estimators of spatial variance ($\hat{\sigma}_s^2$) are not computed, so the bias is not available. On the row-column parametrization (RC) the estimated spatial variance was the sum of the variances attributed to the row and column terms.

The dispersion around genetic variance estimates was nearly constant (Figure 4), as the simulations were based on sampling the heritability from a uniform distribution (0,1). The genetic and spatial variance components were generally biased downwards (Supplementary Tables S7–S10), and the residuals were biased upwards (Supplementary

Tables S11 and S12). Table 1 indicates that genetic variance is more biased downwards when the genetic term is fit without relationship (−0.050) compared to with pedigree (−0.005) and genomic (−0.012) relationship, where pedigree was less biased than genomics. Genetic variances were least biased for spatial parametrization SD and most biased for RC and NS. In marginal terms, the best results were observed from the combination of SD and genomic information (Supplementary Table S8). Figure 4 shows MA and NG providing relatively unbiased genetic variances when associated to pedigree information.

Figure 4 provides three trends related to the bias of spatial variance: (1) in scenarios with deltas of 9:1 and 3:1, the bias of the spatial variance component is lows as the absolute variance is low; (2) models unable to capture the field gradients (NS and RC) display upper biased residual variances, as the variance of spatial trends shifts towards the residuals; (3) SD was the least biased but displayed the highest dispersion.

The unbiasedness of residual variance relies on the accuracy of genetic and spatial terms, as more accurate models yield lower residual variances. Table 1 shows that the bias of residual variance was equally reduced when pedigree (0.142) and genomic relationships (0.141) were utilized in comparison to no relationship (0.185). The spatial parameterizations that provided the least residual bias were SD and AR (Table 1), closely followed by the Gaussian kernel and the Neared Neighbor graph. In one instance, SPDE was combined with genomic information, and the average residual biased was negative −0.003 (Supplementary Table S12). The unbiased SD + G could be an artifact of the simulation settings, as SD is the spatial method closest related to the Matérn function, and both true and statistical models utilize genomic information, so it is expected that results would look considerably different if the true model was based on linear gradients.

## 4. Discussion

Commercial plant breeding programs require making the most out of all information available for decision making, which often entails utilizing sophisticated statistical models that enable the usage of complex inputs, such as genomic relationships and spatial correlations [7,52,57]. Modeling additional sources of information leads to more accurate results and, consequently, better use of the available resources [58,59].

Spatial patterns found in field experiments often constitute a large amount of non-target signals. Ignoring or not properly accounting for exogenous sources of variation causes a reduction in the accuracy of coefficients due to contaminated signals [57,60] and, consequently, inaccurate selections [3,12]. Failing to control and accommodate nuisance parameters is critical when information is scarce [61], which is a common scenario in observational experiments of a non-replicated nature [37,62].

The noisy data problem is critical in the early stages of plant breeding, where decisions have large economic repercussions and are solely based on the information at hand [63]. Normal circumstances dictate that a large number of entries are placed into small plots set under non-replicated experimental design [64]. Plots may be grown primarily for seed increase in early trials; often, breeders perform visual selection relying on checks to discriminate genetic signals from field variation [65]. However, the infusion of information into single-loc evaluations, such as local spatial trends [66] and genomic [67], enables accurate data-driven selections of higher-yielding entries. Besides agronomic performance, models fitting both genomics and spatial information have been beneficial to improve the selection of quality traits, such as grain composition in soybeans [68].

Simulations from this study demonstrate that when there is enough spatial signal in the field, the adequate parameterization of field variation will improve the accuracy of genetic effects under un-replicated trials. In another study with wheat [69], it was reported that trials with replication across locations may also benefit from spatial adjustment. The main findings generally align with previous reports from simulation and real data from various crop and tree species, including studies performed on wheat, rye, barley, triticale, cotton, orange, lupin, tea, and soybeans [3,11,12,58,70–78].

Spatial modeling is a mature statistical approach and is becoming a standard pipeline procedure for analyzing field trials in plant breeding. Autoregressive procedures that account for field variation in agricultural trials were introduced in the 1940s [79,80], with multiple iterations of improvement [2]. Over the past 20 years, AR1 × AR1 became the benchmark model, where the success of this approach is largely due to the sparsity, speed, and availability of its implementation [81]. However, reports indicate some convergence problems with the AR1 × AR1 parameterization under certain conditions, such as its applicability to narrow field layouts [17,62]. This problem has served as motivation to the search for alternative frameworks that may provide equally satisfying results with more algorithmic stability. Previous studies suggested the use of Linear Variance (LV) models [82,83] to improve the stability of AR1xAR1 models, particularly for REML-based implementations (e.g., ASREML, SAS). No convergence problems were observed in this study with the AR1xAR1 implementation fit as a random term with Bayesian estimation correlations and variance components. Even without convergence problems, AR1xAR1 parametrization displayed poor performance under the scenarios with a residual-to-spatial variance ratio lower than 1:1 (Figure 3), which we attributed to the small spatial variance. This issue was not observed in the scenario 1:3 and 1:9, where the spatial variance component is greater than the residual variance. Alternatives to AR1xAR1 commonly involved splines and kernels within the linear model framework [46,78,84,85] as well as new methodologies derived from machine learning, including random forest [86,87] and deep learning [87]. This study utilized a set of methods besides AR1 × AR1, with comparable performance. Similar results were reported [46,83]. We also showed evidence that adding relationship information provided additional benefits to the estimation of both genetics and spatial effects (Table 1).

We find it worth pointing out that, unlike the spatial parameterization, the use of genetic relationship information had not been utilized in plant breeding until recently [10,88], largely motivated by the success of animal breeding and the availability of genomic information. Nowadays, commercial breeding programs have genomic information from every entry grown in the field, and breeding wants to make the most out of this information. A general limitation to the use of genomic information is the cost of genotyping; whereas pedigree information is free, the genomics relationship provided higher accuracies than the pedigree relationship, as supported by numerous reports [26,29,89]. The advantages of genomics over pedigree are believed to be due to the Mendelian segregation and selection bias not captured by pedigree [25,90], and unreliable record-keeping of pedigree information in plant breeding [91,92]. Yet, the pedigree information is a suitable replacement for genomics when such information is not available (Table 1, Figures 3 and 4).

Our simulation settings endeavored to be as realistic as possible. The information utilized in this study is generally available for analysis of field trials in early breeding stages: (1) spatial information is obtained from the field layout map and (2) relationship information is available either through pedigree or genomics, as genotyping is a routine operation in most commercial plant breeding programs. Yet, the advantages of utilizing both sources of information jointly are not of common knowledge, and literature on the topic is scarce. Our simulations provided some evidence that such modeling is beneficial to both genetic effects and parameter estimation (Table 1), leading to more accurate selections in breeding stages where information in the entry basis is not abundant.

We acknowledge that AR1 × AR1, SPDE, and the Gaussian kernel have shared statistical properties with the Matérn function utilized in the true model. As the field variation was simulated as gradients, these conditions are unfavored for row-column adjustments. Thus, terms of rows and columns were not coupled to other gradient-based models. However, row-column adjustment can be as good or better than more sophisticated parameterizations [58], in one specific study with real data.

Gradient-based spatial adjustments are complementary to standard experimental designs [18]. Increments in performance help unbalanced and un-replicated experiments. Among these, the p-rep designs have been emphasized in plant breeding trials in the

early stages [64,93,94]. In such designs, supplementing models with genomic relationship information trends to provide more robust coefficients and variance components without known tradeoffs observed on studies with real and simulated data [21,25,26,95–97]. Theoretical connections between the current work and generalizations to more complex designs and multi-environmental evaluations are provided in the Supplementary Text T1.

Our simulated study provides a new insight on spatial modeling: the residual-to-spatial variance ratio as a determinant factor to define when accuracy gains can be expected from the analysis. Based on the scenarios considered in this study, results indicate that more noise than spatial signal translates into no added benefit on fitting spatial terms.

We envision two directions for future studies: (1) assess and benchmark the benefits provided by fitting spatial and relationship information in multi-environmental trials, with a varying number of locations; (2) evaluate the benefits from adding other sources of information for single-environmental trials, such as genotype-by-environment interactions, dominance, and pleiotropy. The latter might include the simultaneous evaluation of multiple traits, optimization of experimental designs accounting for the number of plots, locations, and phenotyping costs, which become relevant with the advent of phenomics and high-throughput phenotyping.

## 5. Conclusions

The addition of a relationship matrix provided greater improvement to the accuracy of the genetic coefficients than accounting for the spatial variation. The study shows that genomic relationships provide more accurate results than pedigree, and both relationships are better than no relationship.

The modeling field plot variation is beneficial when the spatial variance is equal to or greater than the residual variance. Accounting for field variation using SPDE and Gaussian kernel provided the highest accuracy of genetic effects, closely followed by AR1 $\times$ AR1, Nearest Neighbor Graph, and moving average covariate. In this study, the row-column parametrization was equivalent to no spatial adjustment. We reiterate that our results are conditional to the simulated scenario, as this study emphasized complex field patterns created through the Matérn function.

In terms of parameter estimation, variance components were least biased when relationship information was utilized. Concerning spatial variation, SPDE generally provided the least biased variance components, followed by AR1 $\times$ AR1.

**Supplementary Materials:** The following are available online at https://www.mdpi.com/article/10.3390/agronomy11071397/s1, Table S1: Marginal genetic accuracy and standard error by delta ratios and treatments, Table S2: Marginal genetic accuracy and standard error by relationship and spatial methods, Table S3: Marginal spatial accuracy and standard error by delta ratios and treatments, Table S4: Marginal spatial and standard error by relationship and spatial methods, Table S5: Marginal selection accuracy (top 10%) and standard error by delta ratios and treatments, Table S6: Marginal selection accuracy (top 10%) and standard error by relationship and spatial methods, Table S7: Marginal bias of genetic variances and standard error by delta ratios and treatments, Table S8: Marginal bias of genetic variances and standard error by relationship and spatial methods, Table S9: Marginal bias of spatial variances and standard error by delta ratios and treatments, Table S10: Marginal bias of spatial variances and standard error by relationship and spatial methods, Table S11: Marginal bias of residual variances and standard error by delta ratios and treatments, Table S12: Marginal bias of residual variances and standard error by relationship and spatial methods, Text T1: Factors influencing accuracy within- and across-locations.

**Author Contributions:** É.D.B.d.S. and A.X. conducted the research, wrote the article, designed the simulation and performed data analysis. M.V.F. contributed with the big picture feedback, revised the analysis and interpretation of results. All authors have read and agreed to the published version of the manuscript.

**Funding:** This research received no external funding.

**Institutional Review Board Statement:** Not applicable.

**Informed Consent Statement:** Not applicable.

**Data Availability Statement:** All data utilized in this study was generated through simulation. Software and all simulation parameters are described in the manuscript. The R script is available on GitHub (https://github.com/Ederdbs/SpatialCorrection (accessed on 7 July 2021)).

**Acknowledgments:** Authors thank the Midwest Paraná State University (UNICENTRO) and acknowledge Corteva Agriscience for providing the computational resources to run simulations.

**Conflicts of Interest:** The authors declare that this research was conducted in the absence of any commercial or financial relationships that could be construed as a potential conflict of interest.

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
