# Peer review of "Joint Modeling of Genetics and Field Variation in Plant Breeding Trials Using Relationship and Different Spatial Methods: A Simulation Study of Accuracy and Bias"

_agronomy, doi:10.3390/agronomy11071397_

Round 1

Reviewer 1 Report

The main aim of the paper is to study different genetic parameter regarding plant breeding analysis for spatial ang genetic variance.

The study shows an interesting approach based on simulated data to compare different statistical index to see the one that fits better. The only concern is how to know if the simulated data fits with real data. So discussion should be added about this concern.

Specific comments

Line 68. Explanation of simulation process should be added and references should be given of the accuracy of the data compared with real data.

Line 110. The 900 breeding lines were simulated just once or the process was repeated several time to produce different sets of 900 breeding lines? It would be useful to do it to check the process?

Line 112. This SNPs are commercially available or are only from the simulation process?

Line 366. Any experiment with Soybeans?

Line 419. Real experiments or simulated experiments?

Line 423. Phenotyping cost should be considered, but it is more profitable for the results.

Author Response

Dear Reviewer 1,

We very much appreciate your comments and suggestions, the answers to the questions are attached.

Thanks 

Reviewer 2 Report

report attached

Author Response

Dear Reviewer 2,

We very much appreciate your comments and suggestions, the answers to the questions are attached.

Thanks 

Round 2

Reviewer 2 Report

The inclusion of standard errors in Table 1 suggests that there is no significant difference between some of the methods.  The subsequent rankings should therefore reflect this.

Results from LV analyses should be included in Table 1.

Author Response

The inclusion of standard errors in Table 1 suggests that there is no significant difference between some of the methods.  The subsequent rankings should therefore reflect this.

Now is  reviewed and fixed in paper.

Results from LV analyses should be included in Table 1.

Dear reviewer, we truly appreciate your suggestion. We will recommend the testing of LV for the follow up study. We acknowledge the value it would bring to the publication but we were not able to implement LV in the same solver that has been utilized for all other models, which is the INLA-R.

LV could be a more stable or faster spatial approach than some methods such as splines, RKHS and Gaussian kernel, but we are already providing three models with sparse setup for fast solving, namely: SPDE (sparse precision), AR1xAR1 (sparse precision) and Neighbor Graph (sparse incidence), where the latter has no tuning parameters. All models are providing very similar results, as you pointed out, so we believe that LV (and any other gradient method) would be similar in performance.

Yet, we appreciate the suggestion and we will seriously consider adding LV to the follow up study.